# Combined Cytotoxic Effect of Inhibitors of Proteostasis on Human Colon Cancer Cells

**DOI:** 10.3390/ph15080923

**Published:** 2022-07-25

**Authors:** Alina D. Nikotina, Snezhana A. Vladimirova, Nadezhda E. Kokoreva, Elena Y. Komarova, Nikolay D. Aksenov, Sergey Efremov, Elizaveta Leonova, Rostislav Pavlov, Viktor G. Kartsev, Zhichao Zhang, Boris A. Margulis, Irina V. Guzhova

**Affiliations:** 1Institute of Cytology of Russian Academy of Sciences, Tikhoretsky Ave. 4, 194064 St. Petersburg, Russia; nikotina.ad@gmail.com (A.D.N.); snezhana.alexandrovna@mail.ru (S.A.V.); kokoreva_ne@gkl-kemerovo.ru (N.E.K.); elpouta@yahoo.com (E.Y.K.); aksenovn@gmail.com (N.D.A.); margulis@incras.ru (B.A.M.); 2Saint-Petersburg State University Hospital, Fontanka River enb.154, 190103 St. Petersburg, Russia; sergefremov@mail.ru (S.E.); eliz.leonova@gmail.com (E.L.); rostislavpavlov777@gmail.com (R.P.); 3InterBioScreen, Institutsky Ave. 7a, Chernogolovka, 142432 Moscow, Russia; vkartsev@ibscreen.chg.ru; 4School of Chemistry, Dalian University of Technology, Dalian 116024, China; zczhang@dlut.edu.cn

**Keywords:** colorectal cancer, primary tumor cells, combinatorial therapy, HSF1, Hsp70, chloroquine, CL-43

## Abstract

Despite significant progress in the diagnosis and treatment of colorectal cancer, drug resistance continues to be a major limitation of therapy. In this regard, studies aimed at creating combination therapy are gaining popularity. One of the most promising adjuvants are inhibitors of the proteostasis system, chaperone machinery, and autophagy. The main HSP regulator, HSF1, is overactivated in cancer cells and autophagy sustains the survival of malignant cells. In this work, we focused on the selection of combination therapy for the treatment of rectal cancer cells obtained from patients after tumor biopsy without prior treatment. We characterized the migration, proliferation, and chaperone status in the resulting lines and also found them to be resistant to a number of drugs widely used in the clinic. However, these cells were sensitive to the autophagy inhibitor, chloroquine. For combination therapy, we used an HSF1 activity inhibitor discovered earlier in our laboratory, the cardenolide CL-43, which has already been proven as an auxiliary component of combined therapy in established cell lines. CL-43 effectively suppressed HSF1 activity and Hsp70 expression in all investigated cells. We tested the autophagy inhibitor, chloroquine, in combination with CL-43. Our results indicate that the use of an inhibitor of HSF1 activity in combination with an autophagy inhibitor results in effective cancer cell death, therefore, this therapeutic approach may be a promising treatment regimen for certain patients.

## 1. Introduction

Colorectal cancer is a multifactorial disease with a complex pathogenesis and accounts for approximately 10% of all tumor types [1]. Since 1994, the incidence of colorectal cancer in people under 50 years of age has increased by 2% per year, which is also a cause for concern [2]. From 1950 to 1990, virtually the only drug for the treatment of colorectal cancer was 5-fluorouracil and its derivatives, cytotoxic agents, which are still among the main first-line drugs. Later, other cytotoxic drugs, such as irinotecan (SN-38) and platinum compounds (oxaliplatin), were included in the treatment regimens, and a number of target inhibitors, such as cetuximab, bevacizumab, BRAF inhibitors, and others were discovered somewhat later [3]. Despite the apparent diversity of drugs and treatment regimens, there are a number of problems associated with these approaches. The main disadvantage of using monotherapeutic agents for cancer treatment is tumor heterogeneity, which leads to the selection of resistant cells, in addition to the fact that they act specifically on individual pathways [4]. Furthermore, when cancer cells are exposed to chemotherapy drugs, the surrounding healthy cells and tissues also suffer [5]. In this regard, combination therapy is attracting the attention of an increasing number of researchers. The advantages of combination therapy include the possibility of simultaneously targeting a wide range of oncogenic functions, overcoming drug resistance, and the possibility of a synergistic effect in therapy with several compounds, as well as the use of the drug in low dosages to reduce side effects [6].

The cell proteostasis machinery consists of several major parts aimed at monitoring incorrectly assembled polypeptides and their complexes and includes molecular chaperones, autophagy, and the ubiquitin–proteasomal system of degradation (UPS) [7], which are functionally linked to each other [8]. Proteostasis is the process of regulating intracellular proteins to maintain the balance of the cell proteome, which is crucial for cancer cell survival [9]. Therapy targeted at the proteostasis system can be an effective way to enhance the sensitivity of cancer cells to antitumor therapy.

In cancer cells, the protective function of HSPs and their main regulator, HSF1, contributes to the survival of tumor cells and their metastatic spread [10]. In many tumors of various histogenesis, an increased level of molecular chaperones, particularly Hsp70, is often observed [11]. The main task of Hsp70 in the cell is its protection from physiological, pathological, and environmental influences since it is a key regulator of the protein folding process. Tumor cell survival is critically dependent on the functional activity of the chaperone [12] since it binds to proteins involved in virtually every known process of cell physiology and thus interferes with apoptotic pathways, leading to resistance of cancer cells to anticancer therapy [13].

High autophagy activity is an intrinsic property of highly resistant cells [14], while activation of apoptosis may be considered as a complement to the major therapeutic tool. An antimalarial drug, chloroquine (CQ), has also proven itself in the treatment of a number of tumor diseases [15]. CQ can prevent the fusion of autophagosomes—lysosomes at the initial stages of autophagy—and enhance the antiproliferative effect of chemotherapeutic agents. It has shown sensitizing effects in chemotherapy when used as an adjuvant in cancer clinical trials [16].

The aim of this study was to investigate the efficacy of combinational anticancer therapy using an HSF1 inhibitor found earlier in our laboratory and the autophagy inhibitor, CQ, in colon carcinoma cells taken from cancer patients.

## 2. Results

### 2.1. Tumour Characteristics of Primary Human Colon Carcinoma Cell Lines

Human colon carcinoma (HCC) cell lines HCC6, HCC7, HCC8, and HCC9 were obtained from tumor biopsy materials from untreated patients of the St. Petersburg State University N.I. Pirogov Clinic of High Medical Technologies and transferred into the culture. Cell lines were named according to etiology (HCC) and patient number. According to the degree of malignancy, based on the international classification TxNxMx, the resulting cell lines line up in the following order: HCC8 < HCC6 < HCC9 < HCC7 (Figure 1A). All HCC cells have a similar morphology and cell size (Figure 1B).

In order to determine the cellular characteristics and to understand how their proliferative and migratory abilities could differ, the xCELLigence system was used. We used E-plates or CIM-plates, which allow the recording of a cell index in real time that correlates with proliferative or migratory activity, respectively. The HCC7 line, which is at a higher stage of tumor progression, has a higher proliferative potential than the others, and after 90 h of observation, its cell index reached 3.9 ± 0.3, whereas in the other HCC cell lines, the cellular proliferation index did not exceed values of 2.8 ± 0.4 (Figure 1C); all primary tumor cells had a higher level of proliferative activity than the widely used colorectal cancer line, HCT-15.

A comparison of migration characteristics showed that all cell lines had a similar migration potential, with the exception of HCC6 (IIIB stage), whose cells migrated 1.5 times less intensively. Next, we investigated the sensitivity of the HCC cell lines to antitumor drugs routinely used in the clinic, such as 5-fluorouracil, cisplatin, and etoposide, as well as the subject of current clinical trials, chloroquine (CQ).

Analysis of apoptosis in HCC cells incubated with cisplatin, etoposide, and 5-fluorouracil revealed that only HCC8 cells obtained from non-invasive tumors are sensitive to anticancer drugs, particularly cisplatin; in this case, only 15.3% of the cell population remained alive. Cells of other HCC cell lines demonstrated significant resistance to drugs that are usually used in the clinic (Figure 1D and Appendix A), so we also included CQ on the list CQ which caused significant apoptotic cell death in HCC8, HCC6, and HCC9 cells, whereas in the most malignant line in this study, HCC7 cells, CQ had a modest cytotoxic effect (Figure 1D and Appendix A). We can conclude that cancer cells from primary tumors are predominantly resistant to anticancer drugs.

### 2.2. Hsp70 in Cell Lysate, Cell Media, and on HCC Cell Surfaces

There is much data in the literature indicating a correlation between the intracellular and extracellular levels of Hsp70 and the degree of tumor aggressiveness [13], therefore, we evaluated the amount of (i) intracellular Hsp70, (ii) Hsp70 expressed on the cell surface, and (iii) extracellular Hsp70. To assess intracellular and extracellular Hsp70, we used a recently published Hsp70-ELISA [17]. Using this method, we showed that all HCC cells have high Hsp 70 levels but the most malignant HCC7 (IIIC stage) and HCC6 (IIIB stage) cell lines contain lower amounts of Hsp70 in their cell lysates (CL) and export larger amounts of the chaperone to the cell media. The highest levels of Hsp70 in CL and CM were observed in non-invasive HCC8 cells and HCC9 cells (Figure 2A,B). Importantly, cells of all four lines secreted a high amount of HSP70 into the medium, reaching tens of nanograms, and the variance of this parameter generally corresponded to the amounts of intracellular protein.We did not find any correlation between the degree of malignancy and Hsp70 expression, probably due to the small sample size.

Next, we checked the expression of membrane-bound Hsp70 on the surface of HCC cells. For this purpose, we used FITC-labelled cm.Hsp70.1 monoclonal antibodies [18]. These antibodies were used for flow cytometry experiments and confocal microscopy. According to flow cytometry data, the level of mHsp70 increased with the degree of malignancy (Figure 2C); confocal microscopy data (Figure 2D), and calculation of fluorescence of mHsp70 intensity (488 nm) in every single cell (Figure 2E) confirmed this result. Thus, mHsp70 is more consistent with the grade of malignancy than the intracellular chaperone.

### 2.3. Overcoming the Resistance of HCC Cells to Anticancer Therapy

Since primary tumor cells demonstrated high resistance to antitumor drugs, we considered enhancing their sensitivity with the help of the previously identified HSF1 inhibitor, CL-43 [19,20]. First, we measured the viability of patients’ cells treated with CL-43 in various concentrations using an MTT assay. We found that after 24 h of incubation in a concentration of 10 μM (that is 20-fold higher than the earlier administered dose) 86.4 ± 0.7% of cells survived in the HCC8 population, 70.1 ± 0.3% in HCC6, 80.8 ± 0.3% in HCC9, and 83.4 ± 0.9% in the HCC7 population. The low toxicity of CL-43 is combined with its inhibitory effect on the cell cycle which was illustrated in Appendix A.

Then we used Western blotting of HCC cells incubated with 250 or 500 nM of CL-43 for 20 h with an antibody against HSF1 phosphorylated at S^326^ (phosphorylation of HSF1 residues S^326^ or S^230^ was found to convert HSF1 into a transcriptionally active trimer) [21,22,23] and with an antibody against the HSF1 direct target, Hsp70.

HSF1 activation was decreased in all HCC cell lines by 41–78%; its efficiency was 65%, 78%, 64%, and 41% for HCC8, HCC6, HCC9, and HCC7 respectively. The decrease in Hsp70 expression under the action of CL-43 was more modest and ranged from 13% to 56%. However, one can conclude that CL-43 was quite effective in HCC cells (Figure 3A). To understand whether CL-43 could be toxic to HCC cells, we incubated cells with 500 nM CL-43 and recorded cellular growth using the xCELLigence technique. We found that in HCC8 and HCC7 cells, CL-43 did not cause a significant decrease in the cell index, which may be indicative of mild toxicity, but in HCC6 and HCC9 cells, the use of CL-43 led to a notable decrease in the cell index (Figure 3B). Recently, using the established human colon carcinoma cell line HCT-116, we demonstrated that 500 nM CL-43 led to cell cycle arrest [19]. We checked if this could also be the case for HCC cell lines and found that in HCC6 and HCC9, the proportion of cells in the G2-M phase increased, which was apparently the reason for the additional decrease in the cellular index (Appendix A). 

To understand whether CL-43 suppression of HSF1 activity was sufficient to increase the sensitivity of HCC cells to anticancer drugs, we used combinational therapy of HCC cells with CL-43. For these experiments, we chose CQ. This choice is obvious since we suppress proteostasis by two routes, down-regulation of HSF1 and autophagy. To evaluate the combination effect on HCC cells we used the xCELLigence technique. The data demonstrated that the combination of CL-43 and CQ was more effective; with this drug combination, the cell index in all HCC cells decreased to nearly ‘zero’ (Figure 4A). Since the cell index in HCC7 cells was significantly lower when exposed to CQ compared to untreated cells, but the level of CQ-induced apoptosis was modest (Figure 1D), we hypothesized that CQ could affect the cell cycle. Experimental verification with the help of flow cytometry showed that CQ caused cell cycle arrest in the S phase in all four lines, and in HCC7 cells, the proportion of cells in the S-phase was 85.35% (Figure 4B), therefore, CQ alone predominantly inhibited proliferation in HCC7 cells, whereas the combination of CQ and CL-43 caused cell death in almost the entire population of tumor cells (Figure 4).

Next, we checked whether CL-43 would reduce the resistance of HCC cells to clinically approved anticancer drugs, such as 5-fluorouracil, etoposide, or cisplatin (Figure 1D). In all four HCC lines CL-43 at a concentration of 250 nM did not cause any toxicity as shown with the aid of an MTT assay; the lowest amount of living cells we observed was in HCC8 cells (87.9 ± 1.6). When used in combination with antitumor drugs, the decrease in the proportion of living cells with the use of CL-43 and an appropriate medicine ranged from 20.6% to 49.9% (Figure 4C); this allows us to talk about a synergistic effect of CL-43 with anticancer drugs.

Finally, to make sure that the therapeutic effect obtained for all four primary cell lines extends to other cancer cells of the same histogenesis, we additionally studied the effect of CL-43 in combination with CQ on three established colon cancer cell lines. HCT-15, DLD1, and HCT-116 cells were treated with CL-43 (250 nM) and CQ (60 µM) separately or in combination, and the apoptosis level was determined using flow cytometry. We did not find any CL-43 toxicity compared with untreated cells in all three lines, however, if employed in combination with CQ, CL-43 reduced the proportion of living cells to 3.8% for HCT-15 cells, 11.8% for DLD1, and 44.8% for HCT-116 (Figure 4D).

## 3. Discussion

The system of molecular chaperones, autophagy, and the ubiquitin–proteasome system, one of the main participants in protein homeostasis, are considered potential targets of anticancer therapy, since they affect various processes in a cancer cell, such as proliferation, migration, metastasis, and avoidance of apoptosis [24,25,26].

Our study focused on overcoming drug resistance in colorectal cancer cells using molecules that target these systems in the cell. First, we focused our attention on inhibition of the cell chaperone apparatus, in particular, Hsp70 protein, which, together with HSP90, is considered one of the key players involved in tumor development [27]. They are involved in the regulation of processes such as apoptosis, autophagy, the oncogene-induced aging program, angiogenesis, invasion, and metastasis. At the same time, Hsp70 binds to almost all unfolded or misfolded proteins, while Hsp90 interacts with a specific set of targets [28]. There is ample evidence in the literature of the use of HSP90 inhibitors in various cancer models [29]. However, this approach has its limitations, associated in particular with the compensatory synthesis of Hsp70, which reduces the effectiveness of treatment [30].

Taking into account the importance of Hsp70 to the cancer cell and the data on the correlation between the amount of this protein in the cell and the degree of tumor aggressiveness, we measured the amount of intracellular and membrane-bound Hsp70 and chaperone in the cell medium. However, our results did not confirm the relationship between Hsp70 expression in tumor cells and their malignancy. The highest expression of this protein was observed in HCC8 cells which corresponded to stage II of tumor progression. However, membrane-bound Hsp70 increased with the degree of malignancy. Earlier, it was reported that the amount of Hsp70 in the membrane is dependent on the total cellular amount of Hsp70 [31,32]; however, in our experiments, we observed a correlation between cellular Hsp70 and chaperone in the cell medium but not between cellular and membrane-bound Hsp70. We did not find any dependence on the ability of cells to migrate. Thus, HCC6 (IIIB stage) showed practically no signs of motility, while other lines were highly motile. However, we noted a correlation between the proliferative activity of cells and the stage of tumor progression and with mHsp70 level; the most actively proliferating were the HCC7 cells (IIIC stage, highly invasive). Interestingly, we did not find a strong correlation between drug resistance and Hsp70 levels in cells. The HCC8 and HCC9 lines, which had higher cell Hsp70 levels, demonstrated higher sensitivity to cisplatin and etoposide, whereas HCC6 and HCC9, with low Hsp70 levels, were resistant to these drugs. We found a stronger correlation between the level of membrane-bound Hsp70 and resistance to anticancer drugs in HCC cells. For example, the most malignant HCC7 cells that had the highest level of mHsp70 were highly resistant to all anti-tumor drugs that we used. Hsp70 plays a dual role on the tumor cell surface: mHsp70 serves as a tumor-specific target for natural killer cells [33,34] but can stabilize cell membranes under stress and is cytoprotective against apoptosis-inducing mechanisms [35].

Based on these results, inhibition of Hsp70 alone did not seem to be a promising strategy, so, to overcome tumor cell resistance to anticancer drugs, we used an HSF1 inhibitor that is able to reduce the expression of Hsp90, Hsp70, and Hsp 40 simultaneously [19]. Moreover, HSF1 controls the transcriptional program of other genes that support highly malignant human cancers [36], which makes its suppression even more attractive. For this purpose, we used CL-43, an HSF1 inhibitor discovered earlier in our laboratory [19]. CL-43 belongs to the cardenolide family, but unlike most substances of this class, it possesses mild toxicity. CL-43 diminished HSF1 activity as well as Hsp70 expression in all HCC cells and did not demonstrate significant toxicity; the diminishment of the cell index in HCC6 and HCC9 cells incubated with CL-43 was caused by cell cycle arrest.

For combination therapy with CL-43, we chose CQ for the following reasons: an antimalarial drug, CQ has been used clinically for many years, and over the past decades, it has been established as a potential component of anticancer therapy. Autophagy sustains the survival of malignant cells [37] and CQ inhibits autophagy and induces apoptosis, which has prompted the testing of CQ in various experimental cancer models and human clinical trials. Sensitivity to CQ has been demonstrated in a wide range of tumors, including osteosarcoma, glioblastoma, lung cancer, colorectal cancer, and many others [38,39]. However, in therapeutic schemes presented in the literature, CQ is usually used in combination with routine anticancer drugs, such as 5-fluorouracil, which was effective against colorectal cancer in vitro [40], or cisplatin, where CQ synergistically increased the activity of cisplatin in cisplatin-resistant nasopharyngeal cancer cells [41]. In our experiments, we used two drugs that are intended as ancillary treatments to increase the efficacy of traditional chemotherapy, an inhibitor of the heat shock response and an autophagy inhibitor, and we observed the death of almost the entire population of tumor cells.

## 4. Material and Methods

### 4.1. Cell Culture

Colon carcinoma cells HCC6, HCC7, HCC8, and HCC9 were obtained from tumor biopsy materials from patients of the St. Petersburg State University N.I. Pirogov Clinic of High Medical Technologies in accordance with the rules of the ethics commission. On the same day the biopsies were performed, the tumor tissue was mechanically disintegrated and the cell suspension was seeded to wells of a 12-well plate (TPP, Trasadingen, Switzerland). Cells had three passages and were frozen in liquid nitrogen. For the experiments, the patients’ cells were thawed and multiplied during at least two passages. Cell lines were named according to etiology (Human Colon Carcinoma, HCC) and patient number. HCC9 cells were obtained from biopsy material from a 56-year-old female patient with a diagnosis of colon cancer of the hepatic bend pT3N2b(7/19)M0, G2, PNI-, VI-, IIIC stage [42]; HCC6 cells were obtained from a 69-year-old female patient with a diagnosis of ascending colon cancer pT3N1a(1/26)M0, G3, VI-, PNI-, IIIB stage [42]; HCC7 cells came from a 69-year-old patient female patient with a diagnosis of sigmoid colon cancer (adenocarcinoma) pT3N2b(16/23)M0 G2 PNI+ LV+ IIIC stage [42]; and HCC8 cells were obtained from an 85-year-old male patient with a diagnosis of sigmoid colon cancer pT4aN0M0, G1, PNI-, VI-, IIB stage. According to the degree of malignancy, based on the international classification TxNxMx, the resulting cell lines line up in the following order: HCC8 < HCC6 < HCC9 < HCC7.

On the same day that the biopsy was performed, the tumor tissue was mechanically disintegrated, and the cell suspension was seeded to a 12-well plate (TPP, Trasadingen, Switzerland). The cells had three passages and were then frozen in liquid nitrogen. For the experiments, HCC cells were thawed and multiplied in (at least) two passages. The cells were cultivated in Dulbecco’s Modified Eagle Medium (BioloT, Russia) supplemented with 100 U/mL penicillin and 0.1 mg/mL streptomycin, as well as 10% fetal bovine serum (HyClone, Logan, UT, USA) at 37 °C and 5% CO_2_. As a conditional control (characterized and widespread model of colorectal cancer in culture), HCT15 human colorectal carcinoma cells were selected and cultivated on RPMI 1640 medium (BioloT, Russia) supplemented with 100 u/mL penicillin, 0.1 mg/mL streptomycin, and 10% FBS under the same conditions.

### 4.2. Drugs

Compound CL-43 was obtained from the InterBioScreen collection, dissolved in dimethyl sulfoxide (DMSO) to obtain an initial concentration of 20 mM, and stored at −20 °C until use. Etoposide, doxorubicin, cisplatin, oxaliplatin, chloroquine, PAC-1, and 5-fluorouracil (all from Sigma-Aldrich, St. Louis, MO, USA) were also used in the study.

### 4.3. Analysis of Migration, Proliferation, and Viability of HCC Cells Using the xCELLigence System

The xCELLigence RTCA DP device (ACEA Biosciences, San Diego, CA, USA) was used to assess the cell migration or proliferation characteristics. The advantage of this system is the possibility of cell behavior recording in real-time. Cells (8000 per well) were seeded in E-plates for proliferation analysis or in the upper chamber of CIM-plates in a serum-free medium for the analysis of migration activity. The lower chamber contained a chemoattractant (10% FBS). The chambers are separated by a microporous membrane (pore size approximately 8 µm), the underside of which is covered with gold electrodes. They register the change in resistance during the passage of cells through the pores, which increases with the increase in the number of migrating cells. Each survey was carried out for 50 h; the results were processed using RTCA Data Analysis Software 2.0 (Acea Bio, San Diego, CA, USA).

Cytotoxicity analysis was also performed using the xCELLigence system. HCC cells were placed in 16-well E-plates (ACEA Biosciences, San Diego, CA, USA) at a concentration of 8000 cells/mL and 18 h later were treated with CL-43 at 500 nM alone or in combination with 75 μm chloroquine. Next, the dynamics of cell proliferative activity were recorded for 48 h using the RTCA xCELLigence real-time cell analyzer. Data analysis was performed using RTCA analysis software.

To determine the cytotoxicity caused by the use of antitumor drugs or proteostasis inhibitors in colon cancer cells, we used MTT assay according to the protocol described previously [43].

### 4.4. Hsp70-ELISA Was Employed to Assess the Hsp70 Concentration in HCCs Cell Lysates and Cell Media

Cells were lysed in High RIPA buffer, freeze–thawed three times, ultrasonicated, and centrifuged (5000× *g*). The cell media samples were the supernatants that remained after cell media centrifugation (5000× *g*). To evaluate the Hsp70 content in samples, 100 µL of affinity-purified homemade polyclonal anti-Hsp70 antibody (1.5 mg/mL) was placed in the wells of a 96-well plate (Corning, New York, NY, USA) in 20 mM Borate Suppl buffer (pH 8.0) supplemented with 0.15 M NaCl and incubated for 20 h in a humid chamber. After washing with Buffer A (20 mM Borate buffer [pH 8.0], 0.15 M NaCl, 0.05% Tween-20) the wells were loaded with 100 µL of the appropriate samples as indicated above and with the rHsp70 titers used for calibration; all probes were mixed with 100 µL of Buffer A. Plates were further incubated for 1 h at 37 °C on a shaker and, after washing in Buffer A, biotinylated anti-Hsp70 polyclonal antibody (0.01 µg/mL) was added and the plate incubated for 1 h at 37 °C on a shaker. After washing, StreptAvidin–Peroxidase was added for an additional 1 h. The staining reaction was performed using tetramethylbenzene in citrate buffer (pH 4.5) and the intensity of staining was measured with a Varioscan (Thermo Fisher, Waltham, MA, USA) [17]. All results were obtained in triplicate.

### 4.5. Detection of Apoptosis

Detection of apoptosis was performed with the aid of Annexin V Alexa 647 (Life Technology, Carlsbad, CA, USA) combined with Propidium Iodide staining. HCC cells were treated with one of the following anticancer drugs: etoposide (25 μM), cisplatin (10 μM), 5-fluorouracil (500 μM), or chloroquine (200 μM). After 48 h, cells were collected, washed in cold PBS, resuspended in the binding buffer provided by the manufacturer and stained with Annexin V Alexa 647 and propidium iodide according to the manufacturer’s recommendations. The proportion of cells in each phase of the cell cycle was then determined with the aid of the CytoFlex Flow FACS (Beckman Coulter, Brea, CA, USA) using a laser set at 488 (PI fluorescence) and 638 nm (Alexa647 fluorescence), and the data analyzed with CytExpert 2.0 (Beckman Coulter, Brea, CA, USA) software.

### 4.6. Cell Cycle Analysis

Cells were seeded into a six-well culture plate at 2 × 10^5^ cells/mL and treated with CL-43 or chloroquine. After 36 h of incubation with the compound, harvested cells were washed three times with cold PBS and fixed in 96% ethanol at 4 °C for 20 min. Cells were then treated with 10 μg/mL RNase and stained with 50 μg/mL propidium iodide for 30 min at room temperature in the dark. The proportion of cells in each phase of the cell cycle was then determined with the aid of the CytoFlex Flow FACS (Beckman Coulter, Brea, CA, USA) using a laser with λ = 488 nm.

### 4.7. Confocal Microscopy with Fluorescent Dyes

To visualize the amount of Hsp70 on the cell surface, HCC cells were plated on 24-well plates with pre-loaded coverslips. After they attached to the surface, the cells were washed in PBS and incubated with antibodies to surface HSP70 (Clone 8D1, [17]) on ice. After careful washing, cells were further incubated with secondary antibodies labeled with AlexaFluor555 fluorescent label (Goat anti-Mouse IgG, A28180, Invitrogen, Waltham, MA, USA). Antibodies cmHsp70.1 conjugated with FITC were generously provided by Prof. Gabriele Multhoff, Munich Technical University, Germany. Coverslips with cells were placed in a mounting medium Fluoroshield (Sigma-Aldrich, St. Louis, MO, USA). An Olympus FV3000 confocal laser microscope was used for visualization.

### 4.8. Immunoblotting

To evaluate the expression of HSP70 after treatment with CL-43 at various concentrations, immunoblotting was performed. Human colon carcinoma cells were incubated for 20 h with CL-43 and then washed and lysed in High RIPA buffer (20 mM Tris-HCl pH 7.5, 150 mM NaCl, 0.1% Triton X-100, 0.5% SDS, 1 mM PMSF, 2 mM EDTA, 0.5% DOC). After sonication, lysates were centrifuged at 13,400 rpm. The protein concentration in the supernatant was measured using the Bradford method, and 30 µg of total protein was precipitated with acetone at −20 °C for 20 min and centrifuged at 12,000× *g*; the pellet was dissolved in Laemmli buffer (100 mM Tris-HCl pH 6.8, 4% SDS, 20% glycerol, 0.005% BPS, 50 mM DTT). After electrophoretic separation, proteins were transferred to a nitrocellulose membrane, which was incubated with primary antibodies against HSP70 (clone 3B5) followed by secondary antibodies against mouse immunoglobulins conjugated with horseradish peroxidase (Abcam, Cambridge, UK). Anti-alpha-tubulin antibodies (ThermoFisher, Waltham, MA, USA) were used as the sample loading control.

### 4.9. Statistical Analysis

Numerical results are reported as the mean ± standard error of the mean (SEM) and represent data from three independent experiments. Quantitative analysis was performed with the use of Graph Pad Prism 8.0 (Graph Pad Software Inc, San Diego, CA, USA). One-way ANOVA followed by Dunnett’s multiple comparison test (which compares the mean of each condition with that of the control) was used. Differences were considered statistically significant at *p* < 0.05.

## 5. Conclusions

Based on the results of this work, one can conclude that blocking the proteostasis system leads to the effective death of patient colon cancer cells that are resistant to standard anticancer drugs. The combination of CL-43 and CQ led to almost 100% death of the population of colorectal cancer cells of different cancer stages, which indicates the potential use of this treatment regimen in the clinic. Such a personalized approach may be effective in patients with tumors that are characterized by high expression of Hsp70, HSF1, and autophagy markers.

## Figures and Tables

**Figure 1 pharmaceuticals-15-00923-f001:**
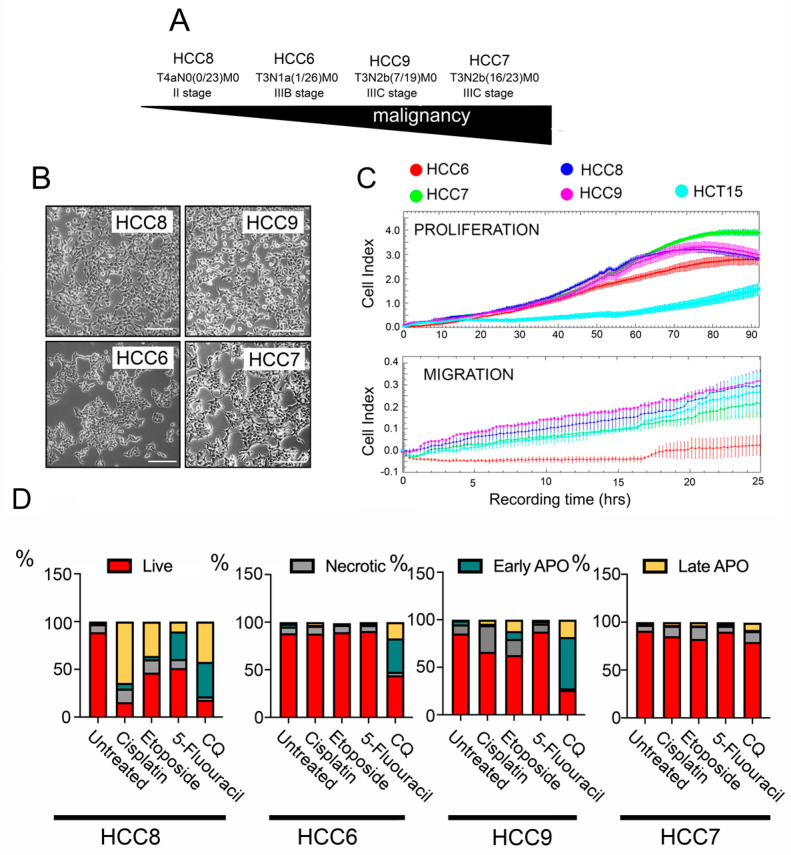
Proliferative migratory properties of human colon carcinoma (HCC) cells and sensitivity to anticancer drugs. (**A**) Arrow demonstrating the level of malignancy of human colon carcinoma (HCC) cells according to medical diagnosis. (**B**) Images demonstrating the morphology of HCC cells. (**C**) Proliferation and migration of HCCs measured with the aid of an xCELLigence real-time cell analyzer. E-plates were used for the proliferation assay shown in the upper graph and CIM-plates for migratory experiments (lower graph). An established cell line of the same histogenesis, HCT-15, was used for standardization. (**D**) Apoptosis in HCC cells incubated with 10 µM of cisplatin, 25 µM of etoposide, 500 µM of 5-fluorouracil, and 200 µM chloroquine for 48 h. The apoptosis level was measured with the aid of flow cytometry. After treatment, cells were stained with Annexin V Alexa 647 and propidium iodide (PI). Flow cytometry diagrams are presented in Appendix A.

**Figure 2 pharmaceuticals-15-00923-f002:**
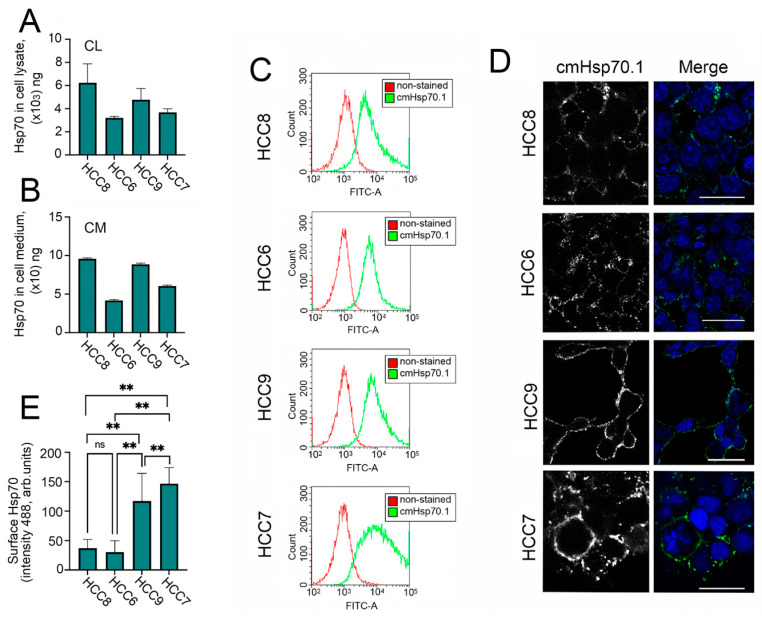
Level of Hsp70 and its localization in human colon carcinoma (HCC) cells, in the cell media, and on the cell surface. (**A**) Level of Hsp70 in HCC cells measured using Hsp70-ELISA calculated as ng/mg of total protein. (**B**) Level of Hsp70 in HCC cell media measured using Hsp70-ELISA and calculated as ng/mL. (**C**) Cell surface Hsp70 determined by flow cytometry (**C**) and by confocal microscopy (**D**). (**E**) Fluorescence intensity of mHsp70 was measured in every single cell with the use ImageJ software. One hundred cells of each cell line were used for analysis. “*ns*”—*non significant*, ** *p* < 0.0001.

**Figure 3 pharmaceuticals-15-00923-f003:**
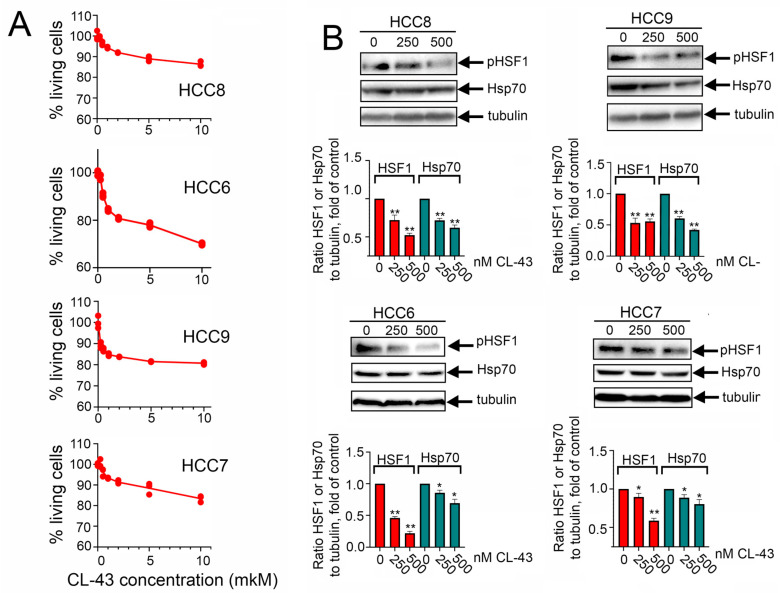
Cardenolide CL-43 effectively decreases HSF1 activity and diminishes the Hsp70 amount in human colon carcinoma (HCC) cells. (**A**) Colon carcinoma cells were seeded to wells of 96-well plates and treated with CL-43 taken in concentrations 0.25, 0.5, 1.0, 2.0, 5.0, and 10.0 μM and after 24 h were subjected to MTT assay. (**B**) Human colon carcinoma cells were incubated with CL-43 in the concentrations indicated (250, 500 nm) and, 20 h later, subjected to Western blotting with antibodies against HSF1^S326^ and Hsp70. Band intensity was measured with the aid of the ImageJ program. Numbers indicate a ratio between HSF1 or Hsp70 band intensity to band intensity of tubulin. * *p* < 0.05; ** *p* < 0.0001.

**Figure 4 pharmaceuticals-15-00923-f004:**
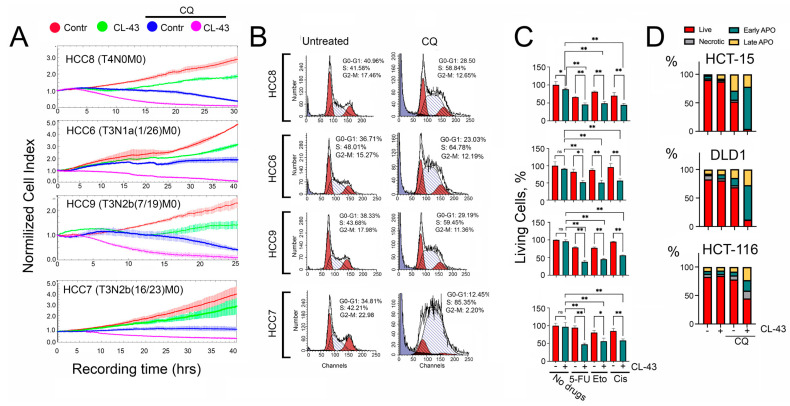
The effectiveness of combination therapy with an HSF1 inhibitor, CL-43, and an autophagy inhibitor, CQ. (**A**) Human colon carcinoma (HCC) cells were seeded onto E-plates and treated with 500 nM of CL-43 in combination with 75 μM of CQ (right column). Recording started immediately after drug administration and lasted 30–40 h. Cell cycle analysis of HCC cells after CL-43 treatment is presented in Appendix A. (**B**) Cell cycle in HCC cells measured 36 h after addition of 75 µM CQ. (**C**) HCC-8,-6,-9,-7 cells were treated with 250 nM CL-43 in combination with 500 µM of 5-fluorouracil, 25 µM etoposide, or 10 µM cisplatin, and 24 h later were subjected to MTT assay. ** p* < 0.05; *** p* < 0.0001. (**D**) Apoptosis in cells of established colon carcinoma cell lines, HCT-15, DLD1, and HCT-116 cells incubated with, 250 nM CL-43 and 60 µM CQ for 48 h was measured with the aid of flow cytometry. After treatment, cells were stained with Annexin V Alexa 647 and propidium iodide (PI). Flow cytometry diagrams are presented in Appendix A.

## Data Availability

HCC cells are available upon request.

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
