# Peer review of "Combined Cytotoxic Effect of Inhibitors of Proteostasis on Human Colon Cancer Cells"

_pharmaceuticals, 2022, doi:10.3390/ph15080923_

Round 1
Reviewer 1 Report
The work is average but may be improved by the inclusion of the following suggestions. It is a case study type of report. This work may be important for some researchers. However, it may be considered for publication by considering the following points.
-Consider the important and related citations (RSC Advances 9, 15357-15369 (2019); Future medicinal chemistry 5 (2), 135-146 (2013); Microbial pathogenesis 53 (2), 66-73 (2012))
-Improving English
-Providing quantitative information.
- Providing concise abstract.
- Providing concise conclusion.
- Providing high quality of Figures.
- Providing error bars in Figures.
Author Response
We are grateful to the reviewer for reading your manuscript and for the criticism that he expressed to. However, some critical points look uncertain. This relates to the demand to cite the papers dealing with the effects of copper(II), nickel(II) and iron(III) complexes of a pyrazoline-based ligand as antitumor and antifungal drugs. These complexes are established as prospect antitumor factors, but our task was to promote the small molecule inhibitors of proteostasis (CL-43 and CQ) which we shown individually or in combination to eradicate most resistant colon cancer cells.
Concerning other comments, our m/s underwent through proof-reading service in UK. We have provided quantitate information.
Reviewer 2 Report
Review of Pharmaceuticals-17522348
This paper reports the use of the cardenolide-type compound CL-43 to inhibit the heat shock protein regulator HSP1, in concert with the autophagy inhibitor chloroquine, to enhance the effectiveness of several cytotoxic drugs such as etoposide, cisplatin, and doxorubicin in a series of human rectal cancer cell lines newly derived from treatment-naïve patients. The cell lines were typed for malignancy and proliferative potential. The results certainly justify their conclusion that blocking the proteostasis system leads to effective cell death in cell lines that are significantly resistant to cytotoxic drugs. Overall the paper is clearly written and laid out, provides abundant data to justify the claims made, and achieves an interesting result.
Author Response
We are grateful to the reviewer for the positive evaluation of our work.
Reviewer 3 Report
In the paper titled "combined cytotoxic effect of inhibitors of the proteostasis on human colon cancer cells” authors established 4 human colon carcinoma cell lines with different malignancy and resistance to drugs. In detail only the less malignant cell lines, namely HCC8 was sensible to cisplatin, etoposide, 5-Fluorouracil, while only the most malignant cell line, namely HCC7 was resistant to CQ treatment. The study focused on overcoming drug resistance in colorectal cancer cells using CL-43 molecule, previously identified as HSF1 inhibitor. Although the topic is interesting for readers, the paper is missing for some experimental validations.
The major concerns are:
1. Authors showed that the decrease of phosphorilated HSF1 was higher for less malignant cells (that are sensible to drugs treatments), and modest for more malignant cells (resistance to drugs treatments). They conclude that CL-43was quite effective in HCC cells. What they mean? Please explain
2. To evaluate whether CL-43 suppression of HSF1 activity was sufficient to increase the sensitivity of HCC cells to anticancer drugs, the combination effect between CL-43 and CQ was detected. It is the reviewer’s opinion that this treatment showed that the combination of drugs is toxic for cells. To demonstrate that the combination of treatment overcome resistance authors should be revert the resistance to cisplatin, etoposide, 5-Fluorouracil. Without this evidence the paragraph overcoming the resistance of HCC cells to anticancer therapy is inconsistent
Minor concerns are:
1. The evaluation of cytotoxicity of CL-43 should be evaluated prior the western blotting analysis
2. Lane 221-222 please adjust. HCC6 and HCC7 contain less HSP in CL (see figure)
3. In figure 2C is not evident the increase of mHsp70 with the increase of malignancy. Please add a graph
Author Response
In the paper titled "combined cytotoxic effect of inhibitors of the proteostasis on human colon cancer cells” authors established 4 human colon carcinoma cell lines with different malignancy and resistance to drugs. In detail only the less malignant cell lines, namely HCC8 was sensible to cisplatin, etoposide, 5-Fluorouracil, while only the most malignant cell line, namely HCC7 was resistant to CQ treatment. The study focused on overcoming drug resistance in colorectal cancer cells using CL-43 molecule, previously identified as HSF1 inhibitor. Although the topic is interesting for readers, the paper is missing for some experimental validations.
We are thankful to reviewer 3 for the attentive reading of our m/s and helpful criticism. We performed new experiments with HCC cells and hope that now the revised version looks more convincing.
The major concerns are:
Q1: Authors showed that the decrease of phosphorilated HSF1 was higher for less malignant cells (that are sensible to drugs treatments), and modest for more malignant cells (resistance to drugs treatments). They conclude that CL-43was quite effective in HCC cells. What they mean? Please explain
A1: We thank the reviewer for this comment. We recalculated reduction of HSF1 activity and rephrased this paragraph which now sounds as it follows: “HSF1 activation was decreased in all HCC cell lines by 78%–41%; its efficiency was 65%, 78%, 64% and 41% for HCC8 and HCC6, HCC9 and HCC7 respectively”. So, redaction of the HSF1 activity with CL-43 was quite effective.
Q2: To evaluate whether CL-43 suppression of HSF1 activity was sufficient to increase the sensitivity of HCC cells to anticancer drugs, the combination effect between CL-43 and CQ was detected. It is the reviewer’s opinion that this treatment showed that the combination of drugs is toxic for cells. To demonstrate that the combination of treatment overcome resistance authors should be revert the resistance to cisplatin, etoposide, 5-Fluorouracil. Without this evidence the paragraph overcoming the resistance of HCC cells to anticancer therapy is inconsistent
A2: We performed additional experiments, using MTT assay and found that CL-43 effectively reduced the resistance of HCC cells to cisplatin, etoposide and 5-Fluorouracil. The data is presented in Figure 4C in new version of the m/s
Minor concerns are:
Q3: The evaluation of cytotoxicity of CL-43 should be evaluated prior the western blotting analysis.
A4: According to results of new experiments performed using MTT assay CL-43 even in concentration 10µM was not toxic for HCC. The data are introduced in Figure 3A prior the western blotting analysis.
Q4: Lane 221-222 please adjust. HCC6 and HCC7 contain less HSP in CL (see figure)
A5: Thank you very much, it was corrected.
Q6: In figure 2C is not evident the increase of mHsp70 with the increase of malignancy. Please add a graph
A6: We calculated the intensity of the fluorescence of mHsp70 using row photos from confocal microscopy. This data is presented in Figure 2E.
Reviewer 4 Report
In this paper, the authors cultured 4 cell lines from colon cancer patients of different stages and compared their drug resistances, migrations and proliferations. They then tested the HSP70 protein levels in different cell lines. After that, the combined therapy of CQ and CL-43 was applied to improve the drug sensitivity. This study applied primary cultured tumor cells as well as some novel technologies. However, the main purpose of the study is just to test the effects of CQ and CL-43 combined effects. In this case, cells from 4 patients is not enough to make the conclusion. Multiple known colon cancer cell lines might be a better choice. Another major issue is that I did not see a significant synergestic inhibitory effect of CQ and CL-43. In Fig. 4, the effect of CL-43 alone should be added. But compared with Fig. 3, the effect is very mild. In summary, the study did not provide significant insight in colon cancer drug resistance. Choice with establised colon cancer cell lines with more mechanistic studies might improve the paper.
Author Response
We would like to thank the reviewer 4 for the helpful criticism, and we hope that the work we have done in response to his/her comments has significantly improved the quality of our manuscript.
In this paper, the authors cultured 4 cell lines from colon cancer patients of different stages and compared their drug resistances, migrations and proliferations. They then tested the HSP70 protein levels in different cell lines. After that, the combined therapy of CQ and CL-43 was applied to improve the drug sensitivity. This study applied primary cultured tumor cells as well as some novel technologies. However, the main purpose of the study is just to test the effects of CQ and CL-43 combined effects. In this case, cells from 4 patients is not enough to make the conclusion. Multiple known colon cancer cell lines might be a better choice. Another major issue is that I did not see a significant synergestic inhibitory effect of CQ and CL-43. In Fig. 4, the effect of CL-43 alone should be added. But compared with Fig. 3, the effect is very mild. In summary, the study did not provide significant insight in colon cancer drug resistance. Choice with establised colon cancer cell lines with more mechanistic studies might improve the paper.
A1: Meeting the reviewers demand we added to research three established colon cancer lines, HCT-15, DLD1 and HCT-116, that were treated with CL-43 and CQ. The result was similar to that we had with HCCs cells. This data is presented in Figure 4D.
A2: We also added the curve of CL-43 to xCELLigence graph. We hope that together with data on established lines synergistic effect looks more obvious.
Round 2
Reviewer 3 Report
The paper has improved and now can be consider for publication.
Reviewer 4 Report
I am OK with the revised version.